# Breast Digital Tomosynthesis versus Contrast-Enhanced Mammography: Comparison of Diagnostic Application and Radiation Dose in a Screening Setting

**DOI:** 10.3390/cancers15092413

**Published:** 2023-04-22

**Authors:** Luca Nicosia, Anna Carla Bozzini, Filippo Pesapane, Anna Rotili, Irene Marinucci, Giulia Signorelli, Samuele Frassoni, Vincenzo Bagnardi, Daniela Origgi, Paolo De Marco, Ida Abiuso, Claudia Sangalli, Nicola Balestreri, Giovanni Corso, Enrico Cassano

**Affiliations:** 1Breast Imaging Division, Radiology Department, IEO European Institute of Oncology IRCCS, 20141 Milan, Italy; 2Department of Statistics and Quantitative Methods, University of Milan-Bicocca, 20126 Milan, Italy; 3Medical Physics Unit, IEO European Institute of Oncology IRCCS, Via Ripamonti 435, 20141 Milan, Italy; 4Radiology Department, Università Degli Studi di Torino, 10124 Turin, Italy; 5Data Management, IEO European Institute of Oncology IRCCS, 20141 Milan, Italy; 6Department of Radiology, IEO European Institute of Oncology IRCCS, 20141 Milan, Italy; 7Division of Breast Surgery, IEO European Institute of Oncology IRCCS, 20141 Milan, Italy; 8Department of Oncology and Hemato-Oncology, University of Milan, 20122 Milan, Italy; 9European Cancer Prevention Organization, 20122 Milan, Italy

**Keywords:** Contrast-Enhanced Mammography, Digital Breast Tomosynthesis, breast cancer screening, Average Glandular Dose

## Abstract

**Simple Summary:**

Screening mammography reduces mortality from breast malignancy. However, breast cancer screening is, unfortunately, hindered due to the poor sensitivity of mammography in dense breasts: up to 15–30% of all cancers may be missed. Given the rapid development of Contrast-Enhanced Mammography (CEM) and its potential for diagnostic use, even in an asymptomatic population, it seems very important to correctly assess the Average Glandular Dose (AGD) for a single CEM examination. Few studies have compared the AGD of CEM versus Digital Mammography (DM) and protocols, including Digital Breast Tomosynthesis (DBT) plus DM, in the same group of patients. The additional role of tomosynthesis versus digital mammography in asymptomatic patients with dense breasts in screening examinations has been well investigated with encouraging results. In this study, we intend to compare the AGD and the diagnostic performance of CEM versus DM, and of CEM versus DM + DBT, performed in the same group of patients over the same period of time in a screening setting.

**Abstract:**

This study aims to evaluate the Average Glandular Dose (AGD) and diagnostic performance of CEM versus Digital Mammography (DM) as well as versus DM plus one-view Digital Breast Tomosynthesis (DBT), which were performed in the same patients at short intervals of time. A preventive screening examination in high-risk asymptomatic patients between 2020 and 2022 was performed with two-view Digital Mammography (DM) projections (Cranio Caudal and Medio Lateral) plus one Digital Breast Tomosynthesis (DBT) projection (mediolateral oblique, MLO) in a single session examination. For all patients in whom we found a suspicious lesion by using DM + DBT, we performed (within two weeks) a CEM examination. AGD and compression force were compared between the diagnostic methods. All lesions identified by DM + DBT were biopsied; then, we assessed whether lesions found by DBT were also highlighted by DM alone and/or by CEM. We enrolled 49 patients with 49 lesions in the study. The median AGD was lower for DM alone than for CEM (3.41 mGy vs. 4.24 mGy, *p* = 0.015). The AGD for CEM was significantly lower than for the DM plus one single projection DBT protocol (4.24 mGy vs. 5.55 mGy, *p* < 0.001). We did not find a statistically significant difference in the median compression force between the CEM and DM + DBT. DM + DBT allows the identification of one more invasive neoplasm one in situ lesion and two high-risk lesions, compared to DM alone. The CEM, compared to DM + DBT, failed to identify only one of the high-risk lesions. According to these results, CEM could be used in the screening of asymptomatic high-risk patients.

## 1. Introduction 

About 12% of the world’s diagnosed neoplasms are breast neoplasms, with about eight million women involved worldwide [1]. It has now been widely recognized that screening mammography reduces mortality from breast malignancy [2]: the mortality rate decreased by about 1.9% annually between 1998 and 2013. 

A screening program aims to find small cancers before they become clinically evident; for example, European guidelines offer mammography every two years for the general female population from 50 to 70 years of age [3]. One of the main issues of mammographic screening is related to the poor sensitivity of mammography in dense breasts; up to 15–30% of all cancers may be missed [4]. 

Some studies have introduced Digital Breast Tomosynthesis (DBT) in mammography screening, especially in high-risk patients with dense breasts [5,6,7]. Those studies found increased cancer detection rates with DBT, from 1.9 to 4.1 per 1000 women screened with recall rates lower than or comparable to those with DM, and an increased cancer detection rate of up to 30–40% [8]. In accordance with the results of these studies, personalized screening based on breast density could be offered that would increase the cancer detection rate and reduce the number of recalls for benign conditions; in fact, DBT seems to be associated, in addition to an increased cancer detection rate, especially in dense breasts, with a higher positive predictive value of recalls [9,10]. 

In this context, moreover, many studies have confirmed the interesting performance of Contrast-Enhanced Mammography (CEM) when applied to the early detection of mammary neoplasms [11], and some preliminary studies have also evaluated the use of CEM in breast-screening programs, with encouraging results [12,13]. These preliminary studies show that CEM has a high negative predictive value in the evaluation of breast lesions and can significantly reduce the number of screening recalls for benign breast conditions, which suggests a potential for a reduction in screening recalls.

To our knowledge, however, the possible role of CEM as an alternative method to conventional screening mammography in some high-risk patient populations has not yet been prospectively investigated.

Given the rapid development of CEM and its potential for diagnostic use, even in an asymptomatic population (screening), it seems very important to correctly assess the Average Glandular Dose (AGD) for a single CEM examination. While many studies have evaluated the diagnostic performance of tomosynthesis and CEM, far fewer studies have compared the AGD of these two methods. In particular, as far as we know, only one study focused on the same group of patients undergoing both tomosynthesis and CEM at short intervals [14]. The dose can be affected by total breast thickness, breast density, and the age of patients [15,16,17].

The purpose of our study is to compare the AGD in the same group of patients undergoing two views of Digital Mammography (DM), plus one view of Digital Breast Tomosynthesis (DBT), followed by Contrast-Enhanced Mammography (CEM) within a short time (within two weeks). The execution of the diagnostic tests we performed on the same patients allowed us to reduce the variances that may occur in different patients (e.g., different breast density, total breast thickness, and age). AGD is a parameter that should be considered before proposing CEM in a screening setting in asymptomatic patients. As a secondary objective, we also wanted to evaluate whether the number of additional breast cancer lesions identified by tomosynthesis (compared with digital mammography alone) was also identified by CEM.

## 2. Materials and Methods

After the approval of our institutional review board and ethics committee, we started to enroll patients for a monocentric and prospective trial. All patients consented by signing a specific, informed consent for the study.

We performed preventive screening examinations in high-risk asymptomatic patients between 2020 and 2022. All participants were offered two-view Digital Mammography (DM) projections (Cranio Caudal and Medio Lateral) plus one projection with Digital Breast Tomosynthesis (DBT) (mediolateral oblique, MLO) in a single session examination; this protocol (DM plus one single projection DBT) reproduces what has already been investigated in the Malmö Breast Tomosynthesis Screening Trial with encouraging results [18].

For all patients in whom we found a suspicious lesion with DM + DBT (B.I.-RADS > 3), according to the Breast Imaging Reporting and Data System [19], a CEM examination (within two weeks) was performed. In each case, we used an AGD protocol in the range of international guidelines [20]. 

Protocol for CEM examination: two bilateral Cranio Caudal (CC) and mediolateral oblique (MLO) projection views were acquired after the intravenous injection of an iodinated contrast agent (Ioexolo) (300 mg/mL, 1.5 mL/kg, Omnipaque^®^, GE Healthcare, Chalfont St. Giles, UK). Two exposures were acquired, one with low energy (26–32 kVp) and one with high energy (45–49 kVp). The low- and high-energy images were then recombined to highlight the uptake of the contrast agent. In none of the cases in our study were late projections acquired.

Three mammography systems (GE^®^ Healthcare, Senographe Pristina^®^, Chalfont St. Giles, UK, or the Amulet^®^ Innovality^®^ Fujifilm, Akasaka Minato-ku Tokyo, Japan or Selenia^®^ Dimension^®^ Hologic, Marlborough, MA, USA) were used for this study.

The device was equipped with these anode/filter combinations: Mo-Mo, Rh/Ag; W/Rh; W/Ag; W/AI; Mo/Cu; Rh/Cu; W/Cu (Mo: molybdenum; Rh: rhodium; Al: aluminum; W: tungsten; Cu: copper; Ag: silver) as we can see in Table 1.

We collected data regarding the age of enrolled patients, breast density, and average glandular dose in Milligray (mGy) of mammographic projections alone (DM), mammographic projections plus Digital Breast Tomosynthesis (DBT), and Contrast-Enhanced Mammography (CEM), respectively. AGD was then compared. We also measured and compared the compression force in Newton (N) and the total breast thickness in millimeters (mm) at the time of examination, both for the DM + DBT protocol and CEM protocol. Doses, in terms of average glandular doses, the compression force and the total breast thickness at the time of examination have been extracted from the DICOM header on images. Average glandular doses reported in the DICOM header were calculated with the Dance model for all the systems [21].

All lesions (BIRADS > 3) identified by DBT were biopsied; we assessed whether lesions found by DBT were also highlighted by DM alone and/or CEM. Two experienced radiologists evaluated all images with more than five years of experience in consensus. (L.N.: 5 years of experience in breast imaging and A.B.: more than 25 years of experience in breast imaging).

Image assessment was performed using the mammographic BIRADS [19] for DM + DBT and the new BIRADS for CEM [22]. In the case of enhancement at CEM, this was evaluated according to the lesion conspicuity descriptor (defined as the enhancement intensity relative to the surrounding background) [22,23].

According to the World Health Organization’s classification of breast tumors, all lesions were categorized depending on the biopsy results [24].

In particular, we considered the following categorizations:B2: benign lesions.B3: high-risk lesionsB5a: in situ lesions.B5b: invasive lesions.

### 2.1. Inclusion Criterion of the Study 

Asymptomatic patients undergoing mammography and tomosynthesis in which there is a dubious finding (BI-RADS > 3).Patients at high risk for the development of breast neoplasia (at least one first-degree relative with breast neoplasm).Patients with suspicious findings (BI-RADS > 3) undergoing CEM examination.Patients who have signed a specific, informed consent for the study.Patients with age > 18 years old.

### 2.2. Exclusion Criterion of the Study 

Patients with known allergy to iodinated contrast medium.Symptomatic patients with palpable breast lumps.Patients with breast implant(s).Patients with proven or supposed pregnancy.

### 2.3. Statistical Analysis

Continuous data were reported as medians and ranges, and categorical data were reported as counts and percentages. The paired *t*-test was used to compare the distribution of the AGD, the total breast thickness, and of the compression force between the examined techniques. A *p*-value less than 0.05 was considered statistically significant. All analyses were performed with the statistical software SAS 9.4 (SAS Institute, Cary, NC, USA).

## 3. Results

From a group of 125 high-risk and asymptomatic patients who have performed mammography plus tomosynthesis, we enrolled 49 patients in the study.

A flowchart diagram of the inclusion/exclusion criteria of the study is shown in Figure 1.

The median age of the patients was 52 years (range: 40–80). In 63% of cases, the lesion found was in microcalcifications, and in 29% of cases was in the form of mass. The breasts were dense in more than 70% of cases, according to the ACR classification [19]. In more than 50% of cases, the BI-RADS classification was 4a.

The descriptive characteristics of the patients are summarized in Table 2.

The median CEM AGD dose of our examinations was 4.24 mGy (range: 1.96–7.60), the median DM AGD was 3.41 mGy (range: 1.14–6.65), and the AGD of DM + one single projection DBT was 5.55 mGy (range: 2.07–10.06) (Figure 2).

The median AGD was lower for DM alone than for CEM (*p* = 0.015), and the AGD dose for CEM was significantly lower than for the DM plus one single projection DBT protocol (*p* < 0.001). The distribution of examinations performed per mammograph was as follows: CEM mammograph: 10 (20%) Fuji Amulet Innovality; 34 (69%) GE Pristina; 5 (10%) Hologic 3Dimensions.DM + DBT mammograph: 26 (53%) GE Pristina; 23 (47%) Hologic 3Dimensions.

Twenty-two patients had the same CEM and DM + DBT mammograph (18 GE Pristina and 4 Hologic 3Dimensions).

Among these 22 patients, the median CEM AGD was 4.51 mGy (range: 3.17–7.60), the median DM AGD alone was 4.31 mGy (range: 2.52–6.65), and the median AGD of DM + one single projection DBT was 6.52 mGy (range: 4.00–10.06) (Figure 3). The results of the paired *t*-test analyses were: CEM vs. DM + DBT: *p* < 0.001; DM vs. DM + DBT: *p* < 0.001; CEM vs. DM: *p* = 0.27.

The median AGD for CEM was significantly lower than for the DM plus one single projection DBT protocol (*p* < 0.001).

The median CEM compression force (N) was 65 (range: 31–114), and the median DM + DBT compression force (N) was 69 (range: 28–116). We did not find a statistically significant difference in the median compression force between the CEM and DM + DBT examinations (*p* = 0.79).

In Figure 4, we can appreciate the similar distribution between compression force in CEM examinations and DM + DBT examinations.

The median CEM breast thickness was 51mm (range: 29–79mm). The median DM + DBT breast thickness was 52 mm, (range: 31–79mm). The *p*-value of a paired *t*-test testing the difference in total breast thickness in the same patients was 0.009.

With the DM + DBT protocol, we identified 49 breast lesions deemed worthy of in-depth biopsy.

The overall histological results of the biopsy are summarized in Table 3. In particular, we obtained 30 (61%) B2 benign lesions, 9 (18%) high-risk B3 lesions, 7 (14%) B5a (in situ lesions), and 3 (6%) B5b (invasive lesions).

The analysis of DM images alone (without evaluating tomosynthesis projection) identified the presence of 38/49 lesions (78%). Of the 11 additional lesions found by DBT image analysis (10 of which were found in very dense breasts), 7 were benign lesions (B2), 2 were high-risk lesions (B3), 1 was an in-situ lesion (B5a), and 1 was an invasive lesion (B5b). In Figure 5, we can appreciate the example of a breast lesion studied with DM, DBT, and CEM.

Figure 5 shows a 41-year-old asymptomatic woman with grade I familiarity with breast neoplasm who performed a preventive oncologic screening examination with our study protocol (DM + DBT).

In (a), we show the left breast studied with DM in the mediolateral projection; here, a barely perceptible breast finding can be appreciated that could be easily interpreted as glandular tissue (arrow). In (b), the same breast is studied with a tomosynthesis acquisition (DBT) that highlights the structural distortion associated with the breast finding (arrow). The recombined CEM image (c) highlights a nodule with suspicious enhancement (arrow).

The analysis of the low-energy CEM image alone, regarding the detection of breast lesions, showed very similar results to the DM analysis (See Table 4). The addition of recombined image analysis (as opposed to low energy analysis alone) allowed us to identify one more in situ lesion and two more invasive neoplasms. A schematic summary of lesion visibility by this method is shown in Table 4 (recombined image and low energy CEM are part of the same method and should therefore be considered together when considering whether there has been lesion detection or not). 

## 4. Discussion 

Early diagnosis of breast cancer is the key to ensuring a better prognosis and management of patients with this disease [25]; mammography is the leading test in breast cancer screening examinations. It has been shown to reduce breast cancer-specific mortality [25]. The diagnostic performance of mammography in breasts with a predominantly adipose component is excellent. However, this performance declines dramatically in predominantly dense and highly dense breasts, representing about 50% of the population [26]. The sensitivity of mammography alone in identifying breast neoplasms in dense breasts may drop 50% below [27]. This value is too low to ensure early diagnosis and adequate management of life-threatening diseases [27,28].

The additional role of tomosynthesis in asymptomatic patients with predominantly dense breasts in screening examinations has been well investigated with encouraging results. DBT has demonstrated an increased cancer detection rate of 2.2 to 2.5 per 1000 screenings [5,6,7,8]. However, one of the main problems associated with the use of tomosynthesis in a screening setting is a non-negligible average glandular dose to which patients are subjected [29]. The possibility of screening examinations in high-risk patients with breast MRI has been investigated with excellent results [30,31]. However, even in this case, some issues must be solved, particularly the high cost, limited availability, and high false-positive rate [32].

Contrast-enhanced mammography has become increasingly popular in recent years, proving to be a technique with excellent diagnostic performance (even in the dense breast) that is relatively fast and cheap, with a low number of false positives [11]. As a result, its use in a screening context of high-risk patients could be a good compromise between cost and benefit. However, few studies have compared the AGD between CEM and tomosynthesis [14,15,16,17]. As far as we know, only one study has compared AGD in the same patients [14]. Our study presents results that are comparable to those of Philips et al. [14] in a group of 45 patients who underwent a CEM protocol and a tomosynthesis protocol at short intervals; in these results, CEM had a significantly lower average glandular dose than mammography protocols performed with tomosynthesis. Furthermore, as discussed in studies that have proposed MRI in high-risk patients [31,32], contrast agent exposure in screening seems to be acceptable for women at 20% or greater lifetime risk of breast cancer [33]. In our opinion, the conclusions reached in MRI on the contrast medium with Gadolinium can also be applied to the iodinated contrast agents used in CEM. One of the most comprehensive literature reviews published by Zanardo et al. [34], analyzed the adverse reactions to the use of contrast agents associated with CEM by evaluating more than 80 studies and using the fixed-effect model. A pooled rate of adverse reactions of 0.82% (0.64–1.05% 95% CI), with a total of 30 adverse reactions in 14,012 patients, was reported. Moreover, of these 30 adverse reactions, only 1 (3%) was found to be severe.

In our study, we reproduced the protocol with tomosynthesis in a single projection: tomosynthesis performed in a single projection lowers the AGD [18]. However, even in this case, according to our experience, CEM demonstrates a lower AGD. Promising results, in terms of the reduction of AGD in tomosynthesis screening protocols, could be obtained using the 2-D synthetic image of tomosynthesis as an alternative to digital mammography [35]. However, many doubts still need to be solved about the diagnostic quality of the 2-D synthetic image of tomosynthesis, which, right now, does not allow it to be considered a safe alternative to digital mammography [36,37].

One of the peculiar aspects of our study is that we performed DM + DBT and CEM in the same patients at short intervals. 

The average glandular dose is, in fact, greatly influenced by the total breast thickness and breast density [20]. Comparing the methods performed on the same patients allowed us to minimize the confounding effect these variables could represent in the comparison. The AGD could be influenced by the different compression forces to which the breast was subjected during the examination; for example, in Figure 4, we can see no statistically significant difference between the average compressive forces to which the breasts were subjected during the examinations. This confounding effect was, therefore, also minimized.

Although a statistically significant difference was found with the paired *t*-test between breast thickness in the same patients and between the two types of examinations (DM + DBT and CEM), the difference was only a few mm (never approaching a centimeter). Therefore, in our opinion, it did not clinically significantly affect the average glandular dose. The variation of only a few millimeters of total breast thickness, in the same patients, between DM + DBT and CEM, can be appreciated in Appendix A (Matched Boxplot).

Finally, in our case history, we found that the CEM analysis identified all the cancerous lesions that tomosynthesis allowed us to identify more than the analysis of digital mammography alone. With the CEM, the two neoplasms would have been identified (one in situ and one infiltrating) that were “missed” with the DM. The results we obtained are therefore encouraging both in terms of AGD and diagnostic utility. In accordance with other preliminary retrospective studies performed on the same topic [38], CEM could be proposed as a screening test, especially in dense breasts, in patients at high risk for the development of neoplastic neoplasia.

One of the main limitations of our study is represented by its monocentric nature and the limited number of patients; therefore, our results should be considered as preliminary results to propose CEM in a screening protocol in asymptomatic, high-risk patients with predominantly dense breasts. Variable in AGD could also depend on system-specific equipment. Unfortunately, in our study, we had few examinations performed with mammograms to compare AGD between different vendors adequately. We recommend that efforts along these lines be made in subsequent studies. For example, based on results available in the literature [39,40], it seems that different vendors expose to different AGDs, depending on total breast thickness.

## 5. Conclusions

Contrast-enhanced mammography offers a lower AGD than digital mammography protocols with added tomosynthesis. Its diagnostic performance is no less than that of digital mammography plus tomosynthesis. Its use in screening settings in dense breasts and high-risk patients appears promising. This work may lay the basis for broader screening studies that confirm these results. CEM could improve detection in high-risk patients with dense breasts compared to conventional screening with a lower dose than protocols, including tomosynthesis. These results, if confirmed, could revolutionize the approach to preventive screening in high-risk patients, especially in cases of dense breasts.

## Figures and Tables

**Figure 1 cancers-15-02413-f001:**
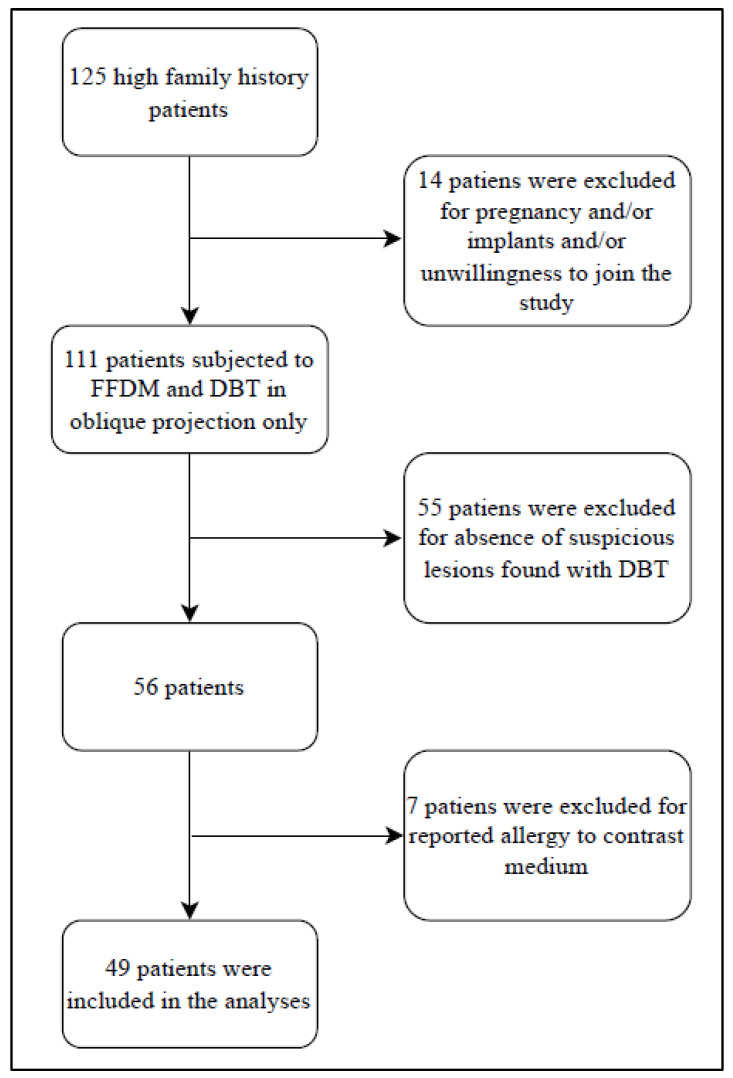
Flowchart diagram of the inclusion and exclusion criteria of the study.

**Figure 2 cancers-15-02413-f002:**
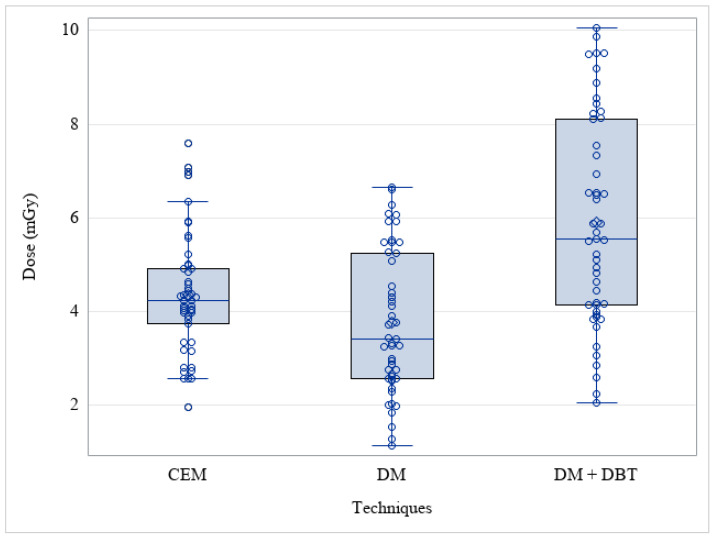
Average Glandular Dose (AGD) among different techniques. (CEM: Contrast-Enhanced Mammography; DM: Digital Mammography; DBT: Digital Breast Tomosynthesis).

**Figure 3 cancers-15-02413-f003:**
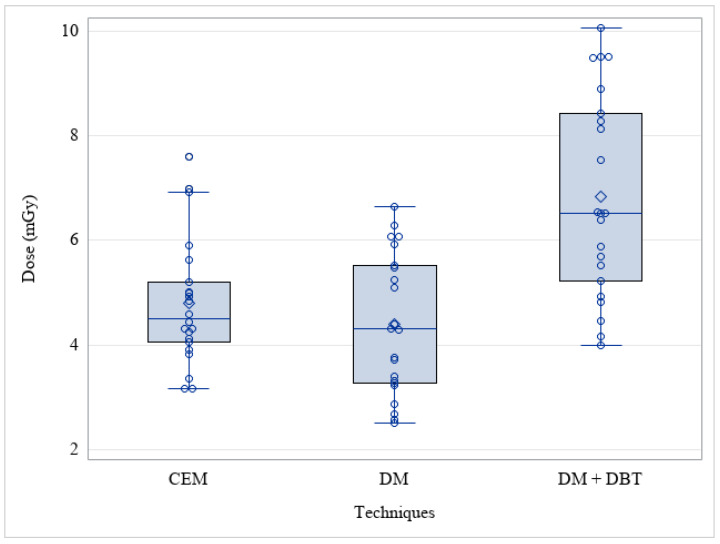
Average glandular dose among different techniques (CEM: Contrast-Enhanced Mammography; DM: Digital Mammography; DBT: Digital Breast Tomosynthesis) among the Twenty-two patients who performed CEM and DM + DBT with the same mammograph.

**Figure 4 cancers-15-02413-f004:**
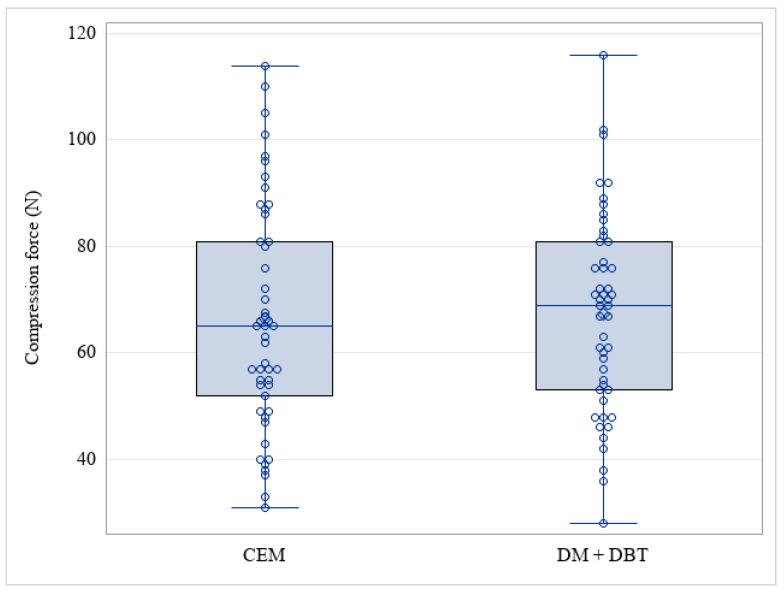
Distribution of CEM and DM + DBT compression force (N) (CEM: Contrast-Enhanced Mammography; DM: Digital Mammography; DBT: Digital Breast Tomosynthesis).

**Figure 5 cancers-15-02413-f005:**
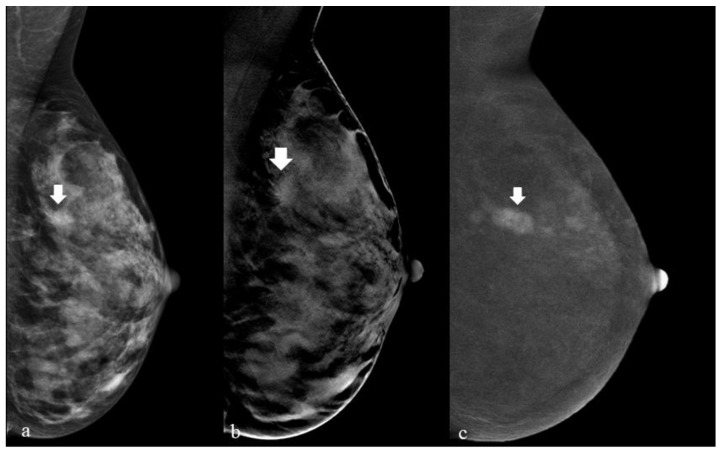
The appearance of a breast lesion with DM, DBT, and CEM.

**Table 1 cancers-15-02413-t001:** Systems used for this study with information of anode/filter combination for Digital Mammography (DM), Digital Breast Tomosynthesis (DBT), and Contrast-Enhanced Mammography (CEM).

System	Anode/Filter DM ^1^	Anode/Filter DBT ^2^	Anode/Filter CEM ^3^ LE ^4^	Anode/Filter CEM HE ^5^
General Electric Pristina	Mo/Mo ^6^ Rh ^7^/Ag	Mo/Mo Rh/Ag	Mo/Mo Rh/Ag	Mo/Cu Rh/Cu
Hologic 3Dimensions	W ^8^/Rh W/Ag ^11^	W/Al ^9^	W/Rh W/Ag	W/Cu ^10^
Fuji Amulet Innovality	W/Rh	W/Al	W/Rh	W/Cu

^1^ DM: Digital Mammography; ^2^ DBT: Digital Breast Tomosynthesis; ^3^ CEM: Contrast-Enhanced mammography; ^4^ LE: low energy; ^5^ HE: High energy; ^6^ Mo: molybdenum; ^7^ Rh: rhodium; ^8^ W: tungsten; ^9^ Al: aluminum; ^10^ Cu: copper; ^11^ Ag: silver.

**Table 2 cancers-15-02413-t002:** Descriptive characteristics of the patients.

Variable	Level	Overall (N = 49)
Age at CEM ^1^ (y ^2^), median (min–max)		52 (40–80)
Type of lesion—DM ^3^ + DBT ^4^, N (%)	Microcalcifications	31 (63)
	Mass	14 (29)
	Mass with microcalcifications	1 (2)
	Architectural distortion	3 (6)
Density (ACR ^5^), N (%)	B	12 (24)
	C	32 (65)
	D	5 (10)
BI-RADS–DBT, N (%)	3	7 (14)
	4a	25 (51)
	4b	8 (16)
	4c	9 (18)

^1^ CEM: Contrast-Enhanced Mammography; ^2^ y: years; ^3^ DM: Digital Mammography; ^4^ DBT: Digital Breast Tomosynthesis; ^5^ ACR: American College of Radiology.

**Table 3 cancers-15-02413-t003:** Histological results of the biopsy (N = 49).

Histological Results of the Biopsy		Overall (N = 49)
B2	Adenosis	4
	Breast fibroadenoma	4
	Ductal hyperplasia without atypia	1
	Fibrocystic disease	18
	Fibrosis	1
	Flogosis	2
B3	Atypical ductal hyperplasia (Din1b)	2
	Atypical lobular hyperplasia (LIN1)	1
	Flat epithelial atypia (FEA)	1
	Intraductal papilloma	4
	Radial scar	1
B5a	High-grade ductal carcinoma in situ (DIN3)	3
	Intermediate-grade ductal carcinoma in situ (DIN2)	2
	Low-grade ductal carcinoma in situ (DIN1c)	2
B5b	Invasive ductal carcinoma	2
	Invasive lobular carcinoma	1

**Table 4 cancers-15-02413-t004:** Histological results of the biopsy and visibility (N = 49).

Histological Results	Visibility									
	DM ^1^		DM + DBT ^2^		Recombined (CEM) ^3^		Low Energy (CEM)		Recombined + Low Energy	
	No	Yes	No	Yes	No	Yes	No	Yes	No	Yes
B2 (N = 30)	7	23	0	30	25	5	8	22	6	24
B3 (N = 9)	2	7	0	9	6	3	2	7	2	7
B5a (N = 7)	1	6	0	7	2	5	1	6	0	7
B5b (N = 3)	1	2	0	3	0	3	2	1	0	3
Total	11	38	0	49	33	16	13	36	8	41

^1^ DM: Digital Mammography; ^2^ DBT: Digital Breast Tomosynthesis; ^3^ CEM: Contrast-Enhanced mammography.

## Data Availability

The data presented in this study are available on request from the corresponding author. The data are not publicly available due to privacy concerns, in accordance with GDPR.

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
