# Peer review of "Breast Digital Tomosynthesis versus Contrast-Enhanced Mammography: Comparison of Diagnostic Application and Radiation Dose in a Screening Setting"

_cancers, 2023, doi:10.3390/cancers15092413_

Round 1
Reviewer 1 Report
This paper provides a valuable contribution to the field of mammography screening by evaluating the dosimetric exposure and diagnostic performance of Contrast-enhanced mammography (CEM) compared to digital mammography (DM) and DM with one-view digital breast tomosynthesis (DBT). The study enrolled 49 high-risk asymptomatic patients and found that CEM offers a lower dose exposure than DM plus DBT protocols while maintaining diagnostic performance. The authors should be commended for their thorough analysis of the data, which included dosimetric exposure and compression force comparisons between the diagnostic methods. The study also assessed whether lesions found by DBT were also highlighted by DM alone and/or CEM. The results showed that the addition of DBT allowed the identification of one invasive neoplasm, one in situ lesion, and two high-risk lesions compared to DM alone.
The findings of this study are particularly promising for the screening of asymptomatic high-risk patients with dense breasts. The authors suggest that CEM could be used as an alternative to DM plus DBT protocols, which often result in higher radiation exposure. Additionally, the study highlights the importance of timely follow-up exams for patients with suspicious lesions, which can be performed within two weeks using CEM. The paired t-test was used to compare the distribution of dosimetric exposure and compression force between the examined techniques. This statistical test is appropriate when the data are normally distributed, and the observations are paired, therefore, the authors should check their data for normality.
The conclusions of this article are extremely short, and should be considerable extended to accommodate the insights on the main outcomes of this study and discuss its implications for the research field.
Overall, this paper provides compelling evidence that CEM is a viable alternative to traditional mammography screening methods with added tomosynthesis. The study's findings have important implications for breast cancer screening, particularly for high-risk patients. The authors' meticulous analysis and interpretation of the data make this paper an important contribution to the field.
Author Response
Plase see attachment.

Reviewer 2 Report
This study provides a useful paired analysis of radiation dose in conventional mammography (DM & DBT) versus contrast-enhanced mammography (CEM) that builds upon a limited body of evidence on this topic.
The value of this paper to the breast imaging community can be increased by several moderate additions and adjustments. I have provided specific comments related to each section below.
Introduction
The brief review of DBT performance could refer to a wider breadth of evidence, especially to convey the variable findings of recall rate differences between DM and DBT that apparently depends on the screening setting and possibly breast density. E.g., https://www.nature.com/articles/s41416-022-01790-x
and https://www.sciencedirect.com/science/article/abs/pii/S1526820922000374
The statement about use of CEM in the screening setting, citing references 10 & 11 is misleading "some preliminary studies have also evaluated the use of CEM in breast 72 screening programs by encouraging results"
Neither of those studies (refs 10,11) used CEM for screening, but rather used it for assessment. Use of CEM for screening certainly looks promising, but to my knowledge, there still aren't any prospective studies to really prove this for a general population. There is good evidence for the diagnostic setting, and for use in various sub-populations.
Methods
The authors report that 3 different systems are used for the study, GE (model needs to be specified), Fuji Amulet Innovality, and the Hologic Selenia Dimensions. However, it isn't clear if the same systems were used for CEM that had been used for the DM/DM+DBT. This should be reported. If more than 1 CEM system was used, then this may complicate the analysis, but I suspect only 1 system was used.
Also, it would be useful to report (e.g., add to Table 1) the breakdown of the proportion of studies acquired on each system type for the reader to understand whether these results may generalize to their setting. This is important as the radiation dose may differ between the different manufacturers and models depending on their technical specifications.
Given compressed breast thickness is the most important determinant of radiation dose (as described by the authors), a comparative analysis of compressed breast thickness should be made. Comparison of compression force alone doesn't rule out potential thickness differences.
Discussion
Please check the reference numbers. Somewhere the link must have been broken as Ref 16 isn't Phillips et al.
The authors suggest that the results pertain to "high-risk patients with dense breasts," but density wasn't part of the inclusion criteria and 12 women with BI-RADS density B were included. Please revise, or expand upon this point to explain.
Overall comment: I would recommend an overall clean-up of the use of terminology related to radiation dose.
Throughout the paper, the use of terminology related to exposure and radiation dose is slightly incorrect here. As it reads now the intent is obvious to the reader so the problem isn't critical to interpretation. But strictly, exposure should refer to the amount of radiation measured in air, or at least to the parameters that are related to the amount of radiation that is incident on the subject. The subjects weren't exposed to a dose, but rather received a radiation dose from exposure to ionising radiation. I think it's probably only a translation issue. But really, the authors aren't comparing 'dosimetric exposure' here, but they are comparing the absorbed dose between the modalities.
E.g., rather than writing "the median exposure dose to radiation" simply say "the median radiation dose."
Reviewer 3 Report
According to the title this paper compares the radiation dose of digital breast tomosynthesis (DBT) and contrast enhanced mammography. However, the materials and methods section is a lengthy description about inclusion and exclusion criteria of a diagnostic study and the in results section only a small part discusses the results of the dose comparison. Instead the types of breast lesions are given extensively and the results of a comparison of compression force between CEM and DM and DBT is given. The latter is not described in the materials and method section.
There is no explanation how the dose values have been obtained. There is little information provided on the systems used in the study (e.g. for GE only the brand is mentioned, not the type) and no information about the X-ray spectra used for DM, DBT and CEM is given, which will influence the dose results. Furthermore it is not clear what is meant with radiation dose. Is this the incident air kerma (entrance dose) or the average glandular dose (AGD)? If it is incident air kerma, this dose depends on X-ray spectrum and therefore does not provide information for the risk of the women. This parameter is not suitable.
If the AGD is given in this study it should be explained how this dose value is calculated and which dose model has been used as this will have major effect on the results. If the results are taken from the DICOM header it should be checked which dose model is used by the different systems.
In the discussion section there are several inaccuracies, I only mention some here:
(1) Terminology: e.g. dosimetric exposure, I guess AGD or incident air kerma is meant or ‘Projection’ in DBT is usually reserved for the individual low dose images made at different angles and not the full DBT exposure.
(2) ‘dosimetric exposure is, in fact, greatly influenced by the thickness of the mammary gland and its density’. Total breast thickness is the major AGD factor, not the thickness of mammary gland.
It is also not clear why it is been stated in the conclusion that ‘the diagnostic performance is no less than that of tomosynthesis’. CEM is used in this study as a follow up exam (if a suspicious finding was found on DM+DBT), so
(1) DBT + DM was used, so the conclusion cannot include solely DBT.
(2) Only lesions visible on DBT+DM are included in the study, lesions only visible on CEM are not included. There is not sufficient information to conclude that CEM is diagnostically equal to DBT+DM.
(3) CEM is used diagnostically as a follow up tool, which is able to provide additional information on tumor neovascularity, so to distinguish benign from malignant structures. It is not meant to replace DBT+DM (except maybe for very dense breasts). The information obtained with DBT+DM is different from the information obtained with CEM, so I do not understand the meaning of ‘diagnostically equal’ in the conclusion.
(4) If CEM is used as screening tool in some trials, this has been used specifically for very dense breast as the mammography technique suffers from low sensitivity for these type of breasts.
(5) In the discussion section the use of contrast agents on a full screening population needs to be discussed.
The authors also do not mention the use of synthetic 2D images instead of ‘real’ 2D images in the discussion section, which will have a major effect on the dose comparison and might lead to different conclusions.
Round 2
Reviewer 2 Report
Thank you for the point-by-point response to my comments. Satisfactory responses were provided for most comments and the quality of the paper has improved overall by the revisions, but some concerns remain.
The revision to the statement about implications of CEM for screening is still too strongly worded. Further revision is suggested. The authors write
“these preliminary studies show that CEM has a high negative predictive value in the evaluation of breast lesions and [can] significantly reduce the number of screening recalls for benign breast conditions”
Here I highlighted the word “can” because those studies didn’t prove that screening recalls could be reduced. The studies were done on screening recalls. As such the results are only suggestive of a potential for a reduction in screening recalls.
The authors’ response to my request for a comparison of compressed breast thickness was not satisfactory. The response was as follows,
“Since the comparison is on the same patients, a short time apart, it can be assumed that breast thickness remains unchanged between the DM+ DBT exam and CEM. Compression force is the only parameter that could have influenced breast thickness in the same patient.”
However, it is simply not true that compression force is the only parameter that could have influenced breast thickness in the same patient. Breast positioning has a substantial impact on the compressed breast thickness, and there is potential for differences in compression thickness between systems for reasons of ergonomics such as the thickness of the detector housing, and/or having more or less rounded edges on that housing. In addition, the use of different compression paddle types such as rigid vs flexible can have an influence.
It is actually quite common for the same patient to have what can be substantial differences (e.g., 1 cm) in compressed breast thicknesses between imaging instances of the same breast in the same view, even when a similar compression force is used. This can even happen during an individual screening exam, such as when a technical repeat is made. The potential for confounders is greater for an exam on a different day, with a different radiographer, and what is potentially a different imaging system, and perhaps a different type of compression paddle.
To avoid all of these confounding effects, one should directly compare the compressed breast thickness rather than use force as a proxy. Given that the authors extracted the force and AGD data from the DICOM header, it shouldn’t be an issue to similarly extract the thickness.
It’s helpful now to learn that multiple different CEM systems were used for this study. While I did appreciate the addition of sub-group analysis for the 22 patients that had imaging on the same system type (could you please provide results of paired t-test for the dose differences here?), it does beg the question as to whether there is variability in AGD depending on system-specific dose levels. I can see that the numbers of exams are too small to reasonably stratify by vendor. However, could you provide some reference dose values that could be compared between systems, such as from phantom testing so that the reader is aware of this confounding effect?
E.g., the Pristina dose has been reported via a phantom test here: https://medphys.royalsurrey.nhs.uk/nccpm/files/other/Tech_Eval_CESMGE_Pristina_NCCPMformatFinalV2.pdf
And similarly for the 3Dimensions: https://medphys.royalsurrey.nhs.uk/nccpm/files/other/Tech_Eval_CESM_Hologic3Dimensions_Final.pdf
The newly added discussion about contrast agent exposure isn’t relevant as the citation, [33], is only applicable to the gadolinium-based agents used in breast MRI. A more appropriate reference should be found.
Author Response
Please see the attachement.

Reviewer 3 Report
The authors have improved the paper and have considered most comments by the reviewer. One larger issue remains and some smaller minor issues.
The minor issues are the inclusion of Tube voltage in the table with x-ray spectra. It is assumed by the reviewer that the authors did a check that all systems used the Dance dosimetry model for the calculation of the glandular dose in the DICOM header. This does not need to be and it cannot de checked in the DICOM header which dose model was used for this calculation.
The large issue which I do not agree with the authors is the term diagnostically equal which occurs e.g. in the conclusions. I believe this term (or non-inferiority) cannot be used as there is no data on false positives for both technologies.
Author Response
The authors have improved the paper and have considered most comments by the reviewer. One larger issue remains and some smaller minor issues.
The minor issues are the inclusion of Tube voltage in the table with x-ray spectra. It is assumed by the reviewer that the authors did a check that all systems used the Dance dosimetry model for the calculation of the glandular dose in the DICOM header. This does not need to be and it cannot de checked in the DICOM header which dose model was used for this calculation.
It’s true, but personal communications with vendors have confirmed the use of the Dance dosimetry model for our systems.
In addition quality dose measurements performed by our medical physicists following the Dance model are in good agreement with the values reported by the systems and in the dicom header.
The large issue which I do not agree with the authors is the term diagnostically equal which occurs e.g. in the conclusions. I believe this term (or non-inferiority) cannot be used as there is no data on false positives for both technologies.
We thank the reviewer for the comment.
We have removed any reference to diagnostic performance and non-inferiority in the text.
We have only used the following terminology: "evaluate whether the number of additional breast cancer lesions identified by tomosynthesis (compared with digital mammography alone) was also identified by CEM."
With this evaluation alone in the text, we are not talking about performance (which would also include the necessary calculation of false positives).
The comparison of performance between the two methods is not the main objective of our study and would not even be correctly set for our study population.

Round 3
Reviewer 2 Report
Thanks for the comprehensive responses to my comments.
I especially appreciate the addition of the thickness evaluation, which I think strengthens the analysis. I appreciate the difference in thickness is small, but now the reader can be convinced. If you are allowed to include supplementary information it might be worthwhile to include the matched boxplot since this is a nice illustration.
My only remaining comment is related to the contrast agent discussion. Rather than the reference [34], I would recommend a CEM-specific reference. In particular, I think you will find this one helpful:
https://insightsimaging.springeropen.com/articles/10.1186/s13244-019-0756-0
This paper specifically surveys adverse reactions while undergoing CEM, which is more relevant data to take into account a typical CEM contrast agent dose. In that review, an adverse reaction rate of 0.82% was found for CEM exams using common contrast agent dose protocols. This should be a relevant rate to take into consideration for a risk/benefit analysis of CEM use.
Author Response
Thanks for the comprehensive responses to my comments.
I especially appreciate the addition of the thickness evaluation, which I think strengthens the analysis. I appreciate the difference in thickness is small, but now the reader can be convinced. If you are allowed to include supplementary information it might be worthwhile to include the matched boxplot since this is a nice illustration.
Thank you for your comment.
As you requested, we have added supplementary figure 1: matched BoxPlot of differences (in millimeters) in total breast thickness, in the same patients, between Contrast Enhanced Mammography and Digital Mammography plus tomosynthesis.
My only remaining comment is related to the contrast agent discussion. Rather than the reference [34], I would recommend a CEM-specific reference. In particular, I think you will find this one helpful:
https://insightsimaging.springeropen.com/articles/10.1186/s13244-019-0756-0
This paper specifically surveys adverse reactions while undergoing CEM, which is more relevant data to take into account a typical CEM contrast agent dose. In that review, an adverse reaction rate of 0.82% was found for CEM exams using common contrast agent dose protocols. This should be a relevant rate to take into consideration for a risk/benefit analysis of CEM use.
Thank you for your comment.
We have replaced reference 34 with the reference you suggested and added some comments to the discussion.
“one of the most comprehensive literature reviews published by Zanardo et al [34], analyzed the adverse reactions to the use of contrast agents associated with CEM by evaluating more than 80 studies and using the fixed-effect model. A pooled rate of adverse reactions of 0.82% (0.64-1.05% 95% CI) with a total of 30 adverse reactions in 14012 patients was reported. Moreover, of these 30 adverse reactions, only 1 (3 %) was found to be severe”.
